# Ultrasound tumor detection using an adapted Mask-RCNN with a continuous objectness score

**Mark Wijkhuizen**[1]                                                    M.WIJKHUIZEN@NKI.NL
**Lennard van Karnenbeek**[1]                                            L.V.KARNENBEEK@NKI.NL
**Freija Geldof**[1]                                                      F.GELDOF@NKI.NL
**Theo Ruers**[1,2]                                                       T.RUERS@NKI.NL
**Behdad Dashtbozorg**[1]                                          B.DASHT.BOZORG@NKI.NL

[1] *Image-Guided Surgery, Netherlands Cancer Institute, Amsterdam, The Netherlands*

[2] *Faculty of Science and Technology, University of Twente, Enschede, The Netherlands*

## Abstract

This paper introduces a novel approach for training Mask-RCNN models using a normalized continuous objectness score and corresponding loss function, eliminating the need for binary objectness labels. The method is evaluated on an ultrasound dataset of breast and colorectal tumors samples, achieving a precision of 0.963, sensitivity of 0.974, specificity of 0.960 and IoU of 0.651, improving the precision and specificity with comparable sensitivity and IoU compared to a conventional Mask-RCNN baseline.

**Keywords:** Ultrasound imaging, Tumor detection, Tumor segmentation, Mask-RCNN

## 1. Introduction

Cancer is the leading cause of death worldwide, causing nearly one in six deaths (Bray et al., 2022). The main treatment for cancer is surgical tumor resection. After the surgery, a pathologist examines the excised tissue for margin assessment. If the margin is positive, there is a high chance of residual tumor tissue in the patient, which requires reintervention or additional treatment. In contrast to pathological margin assessment, an intraoperative margin assessment tool could provide feedback during the surgery. Additional resections can be performed immediately to remove residual tumor tissue, resulting in improvement in patient outcome (Nayyar et al., 2018; Bodilsen et al., 2016).

Ultrasound (US) is a non-invasive and accessible imaging technology that could provide surgeons with intraoperative margin assessment (Hoogteijling et al., 2023), however, the interpretation of US images is challenging and requires training. The latest advancements in machine learning have successfully demonstrated accurate tumor segmentation and margin assessment in US images (Geldof et al., 2023; Veluponnar et al., 2023; Geldof et al., 2022; Tagnamas et al., 2024). However, these models are not optimized for distinguishing healthy frames from frames containing tumor, resulting in false tumor segmentations in the images without any tumors.

Instance segmentation would allow both classification of frames as either healthy or containing tumor and tumor segmentation. A widely used framework for *instance segmentation* is Mask-RCNN (He et al., 2017). This framework is however designed for large datasets with many instances in a single image, not for small medical datasets with a single instance per image.

This paper explores the required adaptations for Mask-RCNN to train on small medical datasets with at most a single instance per image and evaluates this implementation on an in-house breast and colorectal ultrasound dataset. The contributions of this paper are the introduction of a novel normalized objectness score and the corresponding objectness loss function.

## 2. Methods

Mask-RCNN incorporates a region proposal network (RPN) to identify regions of interest and a segmentation head to segment these regions. The RPN predicts an objectness score for each pixel in a feature map grid, where each pixel contains multiple anchors of varying sizes and aspect ratios. Each region proposal above a manually set threshold is further processed in the segmentation head to provide a class and to extract a segmentation mask. All anchors in the RPN have a binary label for a given training sample based on the maximum intersection over union (IoU) with an instance to denote if an anchor contains an instance or background.

The original Mask-RCNN framework (He et al., 2017) assigns a negative label to anchors with an IoU <0.30 and a positive label with an IoU >0.70. The binarization of IoU values originates from the work firstly combining a region proposal and CNN, coined R-CNN. In later iterations, this binarization seems to have been dogmatically applied in Fast R-CNN and Faster R-CNN to eventually being adopted in Mask R-CNN (Girshick et al., 2014)(Girshick, 2015)(Faster, 2015). With large datasets containing many instances per image, there are plenty of positive anchors to train the RPN. However, with small medical ultrasound datasets with a single instance per image, the number of positive anchors per image is minimal. This results in a large class imbalance between background and tumor classes. As a result of this imbalance, the RPN tends to develop a bias towards the background class.

US images are of lower resolution, resulting in a lower-resolution feature map with a larger pixel size. Consequently, neighboring anchors of a positive anchor might have an insufficient IoU to be labeled positive. Additionally, the lower-resolution feature map might also result in samples without a positive anchor due to the misalignment of feature map pixels with the tumor. Thirdly, the single instance per image inherently reduces the number of positive anchors.

In this work, to combat the lack of positive anchors, the binary objectness label is replaced with a continuous objectness score normalized by the maximum global objectness score. With this approach, instead of a few positive anchors, if any at all, each sample has many anchors with an objectness score ranging from 0 to 1. The continuous objectness score is not compatible with the original loss based on a balanced set of anchors. A novel loss function is introduced which takes four subsets of anchors for positive samples and two subsets of anchors for negative samples as shown in Table 1. The total number of anchors $N$ to incorporate in the loss function is a manually set hyper-parameter.

Table 1: Region proposal network loss function for continuous objectness scores.

| Positive frame | Negative Frame |
|---|---|
| • top $\frac{N}{4}$ most positive anchors | • top $\frac{N}{2}$ anchors with highest error |
| • random subset of $\frac{N}{4}$ positive anchors | • random subset of $\frac{N}{2}$ anchors |
| • random subset of $\frac{N}{4}$ negative anchors | |
| • top $\frac{N}{4}$ anchors with highest error | |

An in-house dataset consisting of US frames on both excised breast and colorectal specimens is used to evaluate the training configuration. The dataset is split into a training set of 310 colorectal and 197 breast samples and a test set containing 60 colorectal and 60 breast samples. Both the training and test sets have a 1:1 healthy-to-tumor ratio.

All the US frames are resized to 256×256 before being fed to the EfficientVit-L1 Mask-RCNN backbone(Cai et al., 2023) that produces features maps of scales $\{\frac{256}{4}, \frac{256}{8}, \frac{256}{16}, \frac{256}{32}\}$. The feature maps are processed in a feature pyramid network to create a feature map of size 32×32×32. Two classification heads predict the objectness score and bounding box refinement for each feature map pixel for each scale/size combination. A sample is further processed if the maximum objectness score is above 0.10. Each model is trained for 100 epochs with a batch size of 16 and an Adam optimizer with a peak learning rate of $3^{-4}$ with cosine decay.

## 3. Results & Discussion

All results are averaged over 7 runs with different seeds and show a large gain in precision/specificity with comparable sensitivity and IoU compared to the baseline Mask-RCNN as shown in Table 2. An example of an input frame and corresponding prediction is given in Figure 1. The absence of a public dataset with a balanced healthy-to-tumor sample ratio and research field focus on segmentation create a lack of results to compare with.

Table 2: Quantitative results of the proposed approach compared to the baseline

| Strategy | Precision@10($\sigma$) | Sensitivity@10($\sigma$) | Specificity($\sigma$) | IoU($\sigma$) |
|---|---|---|---|---|
| Baseline | 0.897(0.034) | 0.981(0.006) | 0.882(0.004) | 0.662(0.017) |
| Continuous objectness | 0.946(0.003) | 0.979(0.008) | 0.940(0.004) | 0.654(0.012) |
| Normalized continuous objectness | 0.963(0.028) | 0.974(0.009) | 0.960(0.032) | 0.651(0.019) |

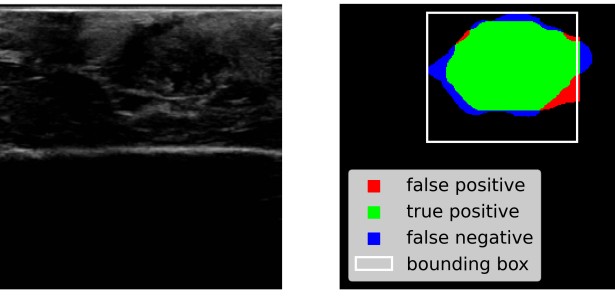

Figure 1: An example of an input ultrasound frame (on left) and corresponding prediction (on right)

Out of 235 training samples containing tumor, 105 would not have any anchor with an IoU above 0.70. With a feature map dimension of 32 and 4 different scales with 3 different ratios this results in roughly 11,000 anchors. Either increasing the feature map dimension or adding additional scales and ratios would further increase the number of anchors, whilst adding only a single positive anchor for these samples in some cases.

A Mask-RCNN model is trained with a novel normalized continuous objectness score and corresponding loss function. This method demonstrates that Mask-RCNN models can be trained without binary objectness labels. Evaluation on an in-house ultrasound dataset of breast and colorectal samples with a balanced healthy-to-tumor ratio shows strong performance with a precision of 0.963, sensitivity of 0.974, specificity of 0.960 and segmentation quality of IoU of 0.651, improving the precision and specificity with comparable sensitivity and IoU compared to a conventional Mask-RCNN baseline.

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
