# OpenReview forum: "Ultrasound tumor detection using an adapted Mask-RCNN with a continuous objectness score"
_MIDL.io/2024/Short_Papers — MIDL 2024 Short Papers_

### Official Review · Reviewer_uTTn · 2024-04-24

**Confidence:** 3
**Final Rating:** 3.5

**Review:**

This paper presents an object detection in 2D ultrasound images of breast and colon using the Mask R-CNN framework.
The clinical context is the use of US intraoperatively, to detect tumor after resection as a sign of suboptimal resection.
Authors propose to formulate the problem as an instance segmentation instead of frame classification (commonly used in state of the art).
They propose a continuous objectness score instead of a binary one, to account for the lack of positive samples in the considered application, but in medical imaging in general as well.
They show that the proposed formulation improves upon baseline Mask RCNN for several performance metrics

#### PROS
* The work originates from the analysis of inner mechanisms of object detection frameworks, from R-CNN to Mask R-CNN, and its impact on medical imaging with few positive samples (with positive anchors)
* The simple proposed solution to replace binary with a continuous objectness score seems to be effective
* Authors propose a novel loos that takes subsets of anchors as positive and negative

#### CONS
* Some details in the paper could have been explained better, for example why 4 and 2 subsets for positive and negative anchors in the proposed loss?
* The paper would have benefitted from more details on data and acquisition.
* Although one of the aims would be to get rid of hyper parameters and thresholds, $N$ remains as a hyperparameter to tune, and a threshold of 0.10 is introduced for the objectness score
* there are no details on who made reference standard
* The work is not reproducible because only private data are used, and no code released, which is a pity as the community might benefit from this modifications across other applications
* It is unclear why Mask R-CNN instead of Faster R-CNN was used for example, performance are reported in term of precision, sensitivity, specificity and IoU, but it is unclear if those are at pixel level for segmentation, at bounding box level, or both. If segmentation is considered, no Dice score is provided.

---

### Decision · Program_Chairs · 2024-04-26

Accept